# Reliability Assessment of Power Systems in High-Load Areas with High Proportion of Gas-Fired Units Considering Natural Gas Loss

Kaile Zeng [1] , Yunchu Wang [1], Shuyang Yu [2], Xinyue Jiang [1], Yuanqian Ma [3], Jien Ma [1] and Zhenzhi Lin [1,*]

[1] College of Electrical Engineering, Zhejiang University, Hangzhou 310027, China; kailez@zju.edu.cn (K.Z.); wangyunchu_ee@zju.edu.cn (Y.W.); weilingjxy@163.com (X.J.); majien@zju.edu.cn (J.M.)
[2] Chaminade College Preparatory, West Hills, CA 91304, USA; syu24@chaminet.org
[3] School of Information Science and Engineering, Zhejiang Sci-Tech University, Hangzhou 310018, China; mayq666@zstu.edu.cn
[*] Correspondence: linzhenzhi@zju.edu.cn

**Abstract:** The "dual-carbon" policy underscores the crucial importance of a secure and stable natural gas supply to ensure the reliable operation of power systems. In high-load areas with a high proportion of gas-fired units and no alternative energy supply, urgent attention needs to be paid to the impact of natural gas loss on power system reliability. Given this background, a method to evaluate power system reliability that considers natural gas supply fluctuations is proposed. In this method, a reliability model of the natural gas supply chain based on the minimal cut set theory is constructed and the influence of policy regulations and economic market factors on system components is quantified. Then, a reliability-evaluation model for a power system that considers gas loss is constructed, and a non-sequential Monte Carlo simulation is used to solve it. Afterward, a reliability-evaluation method considering the power system reserve capacity is proposed. Finally, case studies on a natural gas system with a 14-node power system of a certain area are performed to verify the effectiveness of the proposed method, and the simulation results demonstrate that the reliability of the energy supply directly affects the reliability of the power system.

**Keywords:** natural gas supply chain; power system; minimal cut set theory; non-sequential Monte Carlo simulation; reliability assessment

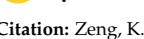



## 1. Introduction

Energy is directly related to social development and economic growth, and its importance is indisputable. Significant changes occurred in the world oil and gas markets in 2022, with political premiums in the oil market, international oil prices rising and then returning to normal levels, and the Brent crude oil annual average price close to $100/barrel. The growth rate of oil demand is lower than expected, and it has not yet reached the level before the pandemic. The increase in oil supply has reached a historic high, and the production reduction of "OPEC+" is significantly higher than planned. Sanctions on Russia have reshaped the crude oil trading pattern, forcing Russia's crude oil exports to shift from the West to the East. Natural gas prices have experienced a historic rise, causing a severe mismatch in energy prices and becoming the initiator of a comprehensive reconstruction of the global energy system [1].

In the context of the Ukraine crisis, various gas prices have doubled and skyrocketed, disrupting the global energy pricing order. Due to Europe's comprehensive disengagement from Russia, under the sharp decline in natural gas demand in Europe, the global natural gas demand has experienced its third decline after 2009 and 2020. European LNG demand has sharply increased, while Russia's pipeline gas exports are restricted, intensifying the global natural gas supply shortage. A major shift has occurred in the natural gas trade

pattern. Taking the largest developing country in the world as an example, natural gas consumption in China is increasing significantly as the country's main energy source. As of the end of June 2022, China's natural gas installed capacity for power generation was 110 million kilowatts, ranking third in the world. It is expected to reach about 150 million kilowatts by 2025 [2]. According to the National Bureau of Statistics, the growth rate of China's coal production in 2022 has slightly slowed down, with a narrowing decline in imports. Coal imports reached 290 million tons, a year-on-year decrease of 9.2%; crude oil production remained stable, with a slowdown in import growth. The imported crude oil was 508.28 million tons, a year-on-year decrease of 0.9%. The growth rate of natural gas production has slowed down slightly. The total natural gas supply in China in 2022 was 370 billion cubic meters. By category, the total increase in domestically produced natural gas was 14.2 billion cubic meters, of which conventional natural gas increased by 9.2 billion cubic meters and shale gas increased by 2.7 billion cubic meters. It is expected that the annual gas consumption of urban residents will reach 143 billion cubic meters, an increase of 10.5 billion cubic meters from 2022. It is expected that the demand for natural gas will continue to grow. The supply and demand situation of primary energy is an important focus for national electricity-related departments and enterprises and other units [3,4].

In the field of the natural gas supply chain, a stochastic model of the natural gas pipeline network capacity was established based on the Markov model and graph theory in [5]. The system capacity was calculated under different scenarios and the reliability of the natural gas supply was assessed. In [6], the natural gas supply chain was combined with the life cycle analysis of gas-fired vehicles to evaluate the impact of methane leakage in the natural gas supply chain. In the field of power system reliability, a scalable Latin hypercube importance sampling method to evaluate power system reliability and reduce the computational load required for introducing renewable energy is proposed in [7]. A power system reliability-evaluation method based on state space segmentation non-repeated sampling is proposed in [8], which ensures a high calculation accuracy and speeds up the convergence rate. A method of evaluating the power system reliability based on deep learning is proposed in [9], taking into account generation and load fluctuations, while ensuring a high calculation accuracy and speed. An improved cost-sensitive assignment method based on the fault severity to evaluate the transient stability of power systems is proposed in [10], which shows a high classification accuracy and excellent generalization ability. A reliability-assessment method for power systems of offshore oil field clusters is proposed in [11], taking into account the production index and providing guidance for actual operation. A reliability model of gas–electric coupling devices to quantify the interdependence of gas–electric systems is established in [12], evaluating the reliability of the distribution system based on the shortest path method. In the field of considering the impact of the natural gas system on the reliability of the power system, a reliability model for the joint operation of electrical and natural gas systems is established and the maximum power that the joint-cycle power plant in the system can provide is calculated in [13]. Fuel availability is considered and a reliability model for the joint operation of power and natural gas systems is established in [14]. However, neither of the above references quantified the impact of gas-transmission losses on the reliability of the power system in a high-load area with a high proportion of gas-fired units.

The researchers above conducted extensive research in the two independent fields of natural gas supply chain and electric power system reliability assessment, but the impact of natural gas supply reduction on the power supply reliability in heavily loaded areas dominated by natural gas is not clear, and there are few relevant studies considering this impact. Therefore, the motivation of this paper is to propose a novel reliability-evaluation method for power systems that accounts for natural gas supply fluctuations and not only considers the impact of the natural gas supply but also considers the influence of external macro factors such as policy regulation and market economic fluctuations on the reliability assessment. The contribution of this paper lies in proposing a method that combines natural gas supply and electric power system reliability assessment for better evaluating

the reliability of electric power systems in heavily loaded areas dominated by natural gas. At the same time, the method proposed in this paper considers the impact of external macro factors, which improves the accuracy and reliability of the evaluation. Firstly, the minimal cut set theory is used to construct the reliability model of the natural gas supply chain, and the influence of policy regulations and economic market factors on system components is quantified. Secondly, a reliability-evaluation model for a power system that considers gas loss is constructed, and a non-sequential Monte Carlo simulation is used to solve it. Thirdly, a reliability-evaluation method considering the power system reserve capacity is proposed. Finally, case studies on a natural gas system with a 14-node power system in a certain area illustrate the effectiveness of the proposed method in evaluating the reliability of the power system considering the natural gas loss.

## 2. Reliability Modeling of the Natural Gas Supply Chain

### 2.1. Natural Gas Supply Chain Reliability Modeling Based on Minimal Cut Set

The natural gas supply chain is composed of gas supply, gas transmission, and gas consumption. The actual natural gas system is a large and complex pipeline network structure formed by various connection methods. Due to the direct relationship between the gas transmission of the pipeline network and its physical topology, using the minimal cut set method to simplify the topology of the gas transmission through the pipeline is proposed in this paper.

The reliability of a single pipeline or a simple pipeline network system is generally calculated by using an intuitive and clear logic block diagram for the established pipeline system model, and such a schematic diagram is shown in Figure 1. In this model, any unit in the minimal cut set is in normal working condition, and when all the component units in the minimal cut set fail, the minimal cut set will fail. This definition of the minimal cut set is associated with the reliability calculation of a parallel system, where the link connection form of the minimal cut set is used as an equivalent parallel structure.

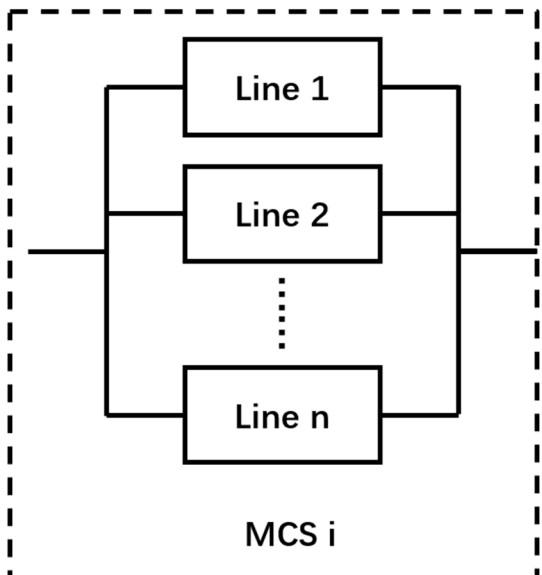

**Figure 1.** Schematic diagram of minimal cut set composition.

The set of minimal events that can lead to the occurrence of the top event is known as the minimal cut set. Any of the minimal cut sets can cause the system to fail, which is why they are connected in a series structure, and the equivalent diagram of it is shown in Figure 2.

In this paper, a method for obtaining the network cut set by identifying the network concatenation set is proposed [15,16]. In the context of the natural gas supply chain, this refers to the pipeline system that connects the gas source to the load at the end of the

transmission. Using graph theory, the gas-transmission path is decomposed and a search tree based on the decomposition results is constructed. The following steps are performed: (1) partitioning and numbering the pipelines in the gas-transmission network; (2) using the gas source point as the starting point and obtaining *n* adjacent gas-transmission lines in the next round of search, which are used as branches for the starting point of the current round; (3) continuing the search from the new branch, ensuring that the search result of the new round is inconsistent with the line number obtained from the previous search before it can be used as a new branch. This step is repeated until the search is completed.

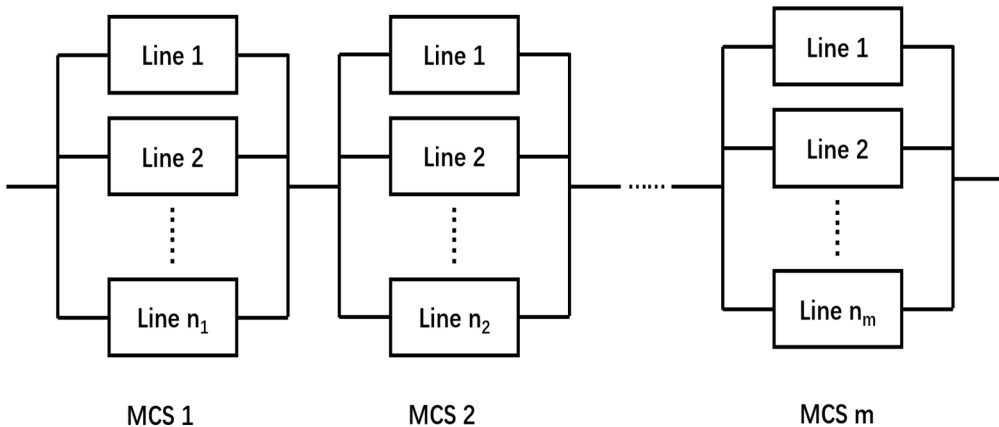

**Figure 2.** Equivalent diagram of a connection mode of minimal cut set.

Due to the limited number of actual natural gas pipeline branches, all paths from the gas source point to the load point can be obtained by traversing the stations (i.e., nodes in the graph) in the supply chain. The search tree from the source point to the load point is a typical recursive structure, which can be represented by the connected set matrix $T$, where the column number of the matrix represents the transmission line number and the number of rows represents the minimum set of connections in the pipeline system. Each row represents a minimum set of connections, and the elements of the matrix $T_{ij}$ are 0–1 variables, where "1" indicates that the pipeline is in the set and "0" indicates that it is not included in the set [17–19].

$$T = \begin{bmatrix} T_{11} & T_{12} & \cdots & T_{1j} \\ T_{21} & T_{22} & \cdots & T_{2j} \\ \vdots & \vdots & \ddots & \vdots \\ T_{i1} & T_{i2} & \cdots & T_{ij} \end{bmatrix} \tag{1}$$

When a column is a unit column vector, it indicates that the gas-transmission lines in this column form the minimal cut set of the network system. If the sum of *n* columns is a unit column vector, it means that they form the *n*-th order minimal cut set of the network system. However, the low-order minimal cut set should not be a subset of the high-order minimal cut set; otherwise, the high-order minimal cut set should be eliminated.

### 2.2. Modeling of Key Influencing Factors in the Natural Gas Supply Chain

The reasons for supply chain dysfunction include insufficient supplier capacity and demand uncertainty on the customer side [20]. The key factors affecting the natural gas supply chain are typically policy regulations and economic market fluctuations, which respectively affect the supply side and the demand side of the natural gas supply chain. The proactive promotion of low-carbon energy transformation policies will reduce the use of heavily polluting energy sources such as coal to a certain extent and increase the proportion of natural gas, leading to further increases in the natural gas supply and demand. However, changes in the international situation [21], such as the Ukrainian crisis, have greatly altered

the natural gas trade pattern. The reduction in the natural gas supply by Russia has caused a tight supply situation, leading to a sharp increase in gas prices in Europe. Similarly, high gas prices have also suppressed the natural gas demand in Asia, leading Southeast Asia to choose other energy sources such as coal and nuclear power as substitutes. Similarly, many other factors such as significant supply and demand fluctuations caused by the COVID-19 pandemic have led to large fluctuations in fossil energy and electricity prices, which have to some extent affected the natural gas supply.

To build an environmentally friendly power system, changes in government policies and regulations can limit the amount of corresponding energy supplied by suppliers, the same as the natural gas supply chain [22]. In addition, when the economic situation fluctuates and the price of natural gas increases, residents may seek alternative resources to replace natural gas, which is equivalent to reducing the supply of gas from the gas source in the natural gas pipeline system. When market policies are regulated and the economic environment is volatile, the gas supply from the source $S$ can be multiplied by a factor $\kappa$, expressed as $S' = \kappa \times S$.

The compressor in a natural gas pipeline system serves a similar function to a step-up transformer in a power grid by raising the pressure level at transmission nodes in the gas network to compensate for pressure loss [23]. However, compressors consume 3% to 5% of the flow passing through them [24]. It is assumed that the natural gas flow is balanced at the compressor under normal conditions, with no loss or increase in flow. If a compressor fails during operation, the gas-transmission capacity of the pipeline connected to the station site is considered to decrease due to changes in the air pressure difference between upstream and downstream pipelines directly connected to it. The degree of decrease is determined by the degree of damage to the compressor set. The sub-station enables filtration, separation of gas impurities, and heating of the gas. If the sub-station is damaged, the quality of natural gas on the customer side will be reduced and the effective gas volume greatly reduced, leading to a decrease in the gas-transmission capacity of the pipeline [25,26]. Therefore, if damage occurs in the pressure station, the pressure sub-transmission station, or the sub-transmission station, the gas-transmission capacity $S$ will be decreased from $\gamma_c\%$, $\gamma_{cd}\%$ and $\gamma_d\%$ to $S'$, denoted as $S' = (1 - \gamma_i\%) \times S$.

As for the pipelines in the gas system, when a rupture or leak occurs in the natural gas pipeline network, it is assumed that the pipeline will not be able to complete its transmission task, and the pipeline's capacity at that point can be expressed as $S' = 0$.

Therefore, the modeling of the impact of key factors on each link of the supply chain is shown in Figure 3 [27], where $\lambda$ is a 0–1 variable, with $\lambda = 1$ representing the failure of each link, and $\lambda = 0$ representing a normal operation of each link [28].

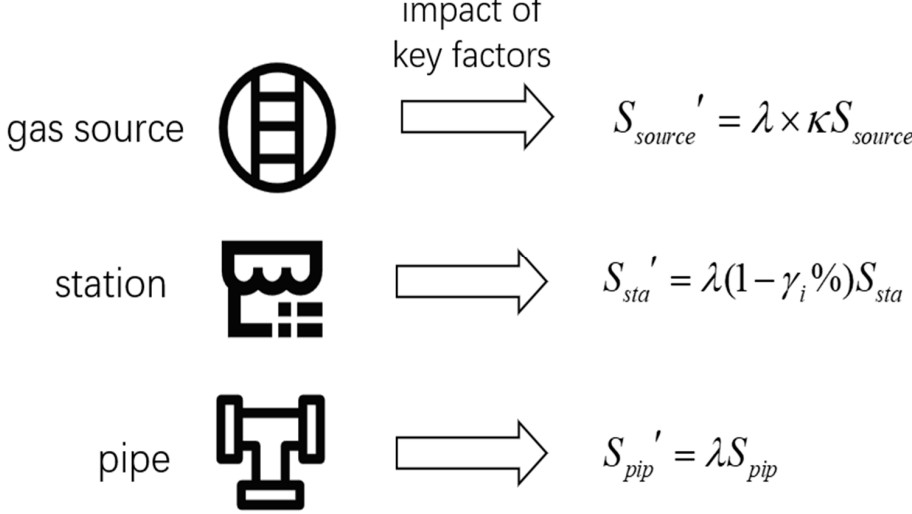

**Figure 3.** Modeling the influence of key factors on gas network components.

### 3. Power System Reliability Assessment Considering Natural Gas Supply Fluctuations

*3.1. Assessment of Gas Supply Capacity of Natural Gas Pipeline System Based on the Monte Carlo Simulation Method*

The natural gas supply chain is a complex system, where a single- or multiple-component failure can potentially cause gas loss events in the supply chain. Exhaustive simulation of all failures through brute force is computationally intensive. Monte Carlo simulation, on the other hand, can solve the problem of dimension explosion caused by brute force and can describe the characteristics of random events and physical processes. Therefore, Monte Carlo simulation is adopted to simulate and calculate natural gas supply failures [29]. The system component states are simulated using the non-sequential Monte Carlo simulation method, which determines each component's state in the system based on the sampling of the component probability distribution and combines each component state as the natural gas supply chain system state. For component $k$, $p_k$ represents the probability of the component's failure [30]. A random number $\xi_k$ is generated in the interval [0, 1]; when $\xi_k \geq p_k$, $S_k = 0$, which means the component is in the operational state. Otherwise, when $0 < \xi_k \leq p_k$, $S_k = 1$, which means the component is in the failed state.

The gas-transmission capacity in the case of system failure is calculated based on the result of the minimal cut set division. Each pipeline in the natural gas system has an upper limit of transmission capacity and a certain transmission margin, i.e., reserve capacity. The reliability of the complex pipeline network system is modeled based on the minimal cut set theory, where each minimal cut set consists of gas-transmission lines in parallel. If a gas-transmission line fails, the gas loss of that line can be compensated for by increasing the gas-transmission capacity of other lines in the same cut set, and vice versa.

Next, the amount of gas lost for the minimal cut set $r$ is calculated. Each line has a maximum allowable gas-transmission volume and an ongoing gas-transmission task, and the difference between the two is the reserve capacity of the line [31]. The reserve capacity of the minimal cut set is equal to the sum of the reserve capacities of all the gas-transmission lines that make up the minimal cut set. The line $l$ is divided into two categories, $l_1$ and $l_2$, where $l_1$ represents lines that occur in more than one minimal cut set and $l_2$ represents lines that occur in only one cut set $r$.

If the spare capacity in the minimal cut set $r$ is less than the lost gas volume of all lines in that cut set, the gas volume of that line cannot be fully compensated. Therefore, the gas loss of line $l$ in cut set $r$ needs to be proportionally allocated according to its share of the lost gas volume of all lines that failed. The gas loss of the line $l$ in cut set $r$ can be calculated and expressed as

$$L_{r,l,i}^{aft} = \left( \sum_{l \in T_r} L_{l,i}^{bef} - B_r \right) \cdot \frac{L_{l,i}^{bef}}{\sum_{l \in T_r} L_{l,i}^{bef}} \tag{2}$$

$$B_r = \sum_{l \in T_r} C_l - \sum_{l \in T_r} S_l \tag{3}$$

where $T_r$ is the set of all lines in the minimal cut set $r$, $B_r$ is the spare capacity of cut set $r$, $C_l$ is the maximum allowable transmission capacity of line $l$, $S_l$ is the ongoing transmission task of line $l$, $L_{r,l,i}^{aft}$ is the calculated transmission loss contribution of line $l$ in cut set $r$ at the $i$-th simulation after proportional allocation, and $L_{l,i}^{bef}$ is the transmission loss due to component failure of line $l$ before proportional allocation at the $i$-th simulation.

All minimal cut sets are traversed, and according to the short-board effect, the minimal cut set with the smallest spare capacity determines the amount of gas loss of the line [32,33]. The gas loss of line $l$ is equal to its maximum gas loss contribution value within all cut sets. After determining the gas loss of the line, the cut set with the larger spare capacity $F_{l_1,i}^{aft}$ is

reallocated to the line gas loss [34], and the gas loss of the remaining lines $F_{l_2,i}^{aft}$ is calculated proportionally for each line in turn, which can be respectively expressed as

$$F_{l_1,i}^{aft} = \max\left\{ L_{1,l_1,i}^{aft}, L_{2,l_1,i}^{aft}, \dots, L_{R,l_1,i}^{aft} \right\} \tag{4}$$

$$F_{l_2,i}^{aft} = \left( \sum_{l_2 \in C_r} L_{l_2,i}^{bef} + \sum_{l_1 \in E_r} F_{l_1,i}^{aft} - B_r \right) \cdot \frac{L_{l_2,i}^{bef}}{\sum_{l_2 \in C_r} L_{l_2,i}^{bef}} \tag{5}$$

where $F_{l_1,i}^{aft}$ is the finalized gas loss of line $l_1$ in the $i$-th simulation after proportional allocation, $R$ is the number of minimal cut sets, $F_{l_2,i}^{aft}$ is the finalized gas loss of line $l_2$ in simulation $i$ after proportional allocation, $L_{l_2,i}^{bef}$ represents the gas loss of line $l_2$ in the $i$-th simulation due to component failure before proportional allocation, $C_r$ is the set of all lines that occur only in cut set $r$, and $E_r$ is the set of all lines in cut set $r$ that also appear in other cut sets.

Multiple Monte Carlo simulations are then performed to obtain the actual volume of natural gas delivered by the pipeline system [35], which can be used as input to the power system, equal to $\left( \sum_{l \in T_r} S_l - \sum_{l \in T_r} F_l^{aft} \right)$.

*3.2. Power System Reliability Assessment Considering Gas Losses*

As a crucial part of the primary energy natural gas supply chain, the electric power system is located on the load side of the natural gas pipeline network supply chain [36]. This paper focuses on studying areas with a high gas-to-machine ratio, no external energy transmission, and high electricity load to evaluate the power system's reliability by considering the gas loss volume of the natural gas pipeline network as the input of the power system under various macro factors such as policy regulations and economic market fluctuations [37].

As the power-generation side of the power system comprises coal-fired units and gas-fired units, while the gas fluctuation event occurs the output of coal-fired units could be increased or part of the load could be cut to maintain the power balance and alter the power flow. Each gas-transmission line of the natural gas pipeline network corresponds to one gas unit on the power system side. The initial state of the gas-transmission line $l$ at node $i$ of the power system and the gas reduction at node $i$ when the gas fluctuation event $w_g$ occurs can be respectively expressed as

$$W_{i,g}^0 = S_l \tag{6}$$

$$\Delta W_{i,g}^{w_g} = F_l^{aft} \tag{7}$$

The above calculations can be used to determine the final power generated by each gas unit, and after linearizing the gas unit's generation model leads, it can be obtained as

$$P_{i,g}^{w_g} = (W_{i,g}^0 - \Delta W_{i,g}^{w_g}) \cdot h_g \cdot \eta_g \tag{8}$$

where $P_{i,g}^{w_g}$ is the power generated by the gas unit at node $i$ during the gas supply fluctuation event $w_g$, $h_g$ is the calorific value of natural gas, and $\eta_g$ is the power-generation efficiency of the gas unit.

Thus, the reserve capacity of the power system can be expressed as

$$G_{sp} = \sum_{i=1}^{N_s} (P_{i,c}^{max} + P_{i,g}^{w_g} - D_{i,L}^0) \tag{9}$$

where $P_{i,c}^{\max}$ is the maximum power generation of coal-fired units at node $i$ in the power system, $P_{i,g}^{w_g}$ is the power generation of gas-fired units at node $i$ during the gas supply fluctuation event $w_g$, $D_{i,L}^0$ is the power load at node $i$ in the initial state, and $N_s$ is the number of nodes in the power system.

According to the above equation, if $G_{sp} \geq 0$, it indicates that the power system has enough reserve capacity to achieve power balance by increasing the power output of each coal-fired unit proportionally [38]. Therefore, after the gas supply fluctuation event $w_g$ occurs, the output power of a coal-fired unit at node $i$ is equal to $P_{i,c}^{w_g}$, and the power load $D_{i,L}^{w_g}$ can be respectively expressed as

$$P_{i,c}^{w_g} = P_{i,c}^0 + \sum_{i=1}^{N_s} (D_{i,L}^0 - P_{i,c}^0 - P_{i,g}^{w_g}) \cdot \frac{(P_{i,c}^{\max} - P_{i,c}^0)}{\sum\limits_{i=1}^{N_s} (P_{i,c}^{\max} - P_{i,c}^0)} \tag{10}$$

$$D_{i,L}^{w_g} = D_{i,L}^0 \tag{11}$$

where $P_{i,c}^0$ denotes the power generated by the coal-fired unit at the initial state of node $i$.

Otherwise, when $G_{sp} < 0$, it means that the reserve capacity of the power system is insufficient at this time, and the power balance of the system cannot be achieved by increasing the output power of coal-fired units alone. The adjustment strategy at this point is to increase the output power of all coal-fired units to their maximum capacity and proportionally reduce part of the electric load [39]. Therefore, after the gas supply fluctuation event $w_g$ occurs, the output power of the coal-fired unit at node $i$, which is expressed as $P_{i,c}^{w_g}$, and the power load $D_{i,L}^{w_g}$ can be respectively expressed as

$$P_{i,c}^{w_g} = P_{i,c}^{\max} \tag{12}$$

$$D_{i,L}^{w_g} = D_{i,L}^0 \cdot \left[ 1 - \frac{\sum\limits_{i=1}^{N_s} (D_{i,L}^0 - P_{i,c}^{\max} - P_{i,g}^{w_g})}{\sum\limits_{i=1}^{N_s} D_{i,L}^0} \right] \tag{13}$$

To evaluate the impact of the natural gas supply on the power system, the following reliability-assessment indicators of the power system are selected, including Expected Demand Not Supplied (*EDNS*), Severity Index (*SI*), and Service Availability (*SA*) [40,41].

Expected Demand Not Supplied *EDNS*: Indicating the amount of power load lost from the system in MW in the event of a gas supply fluctuation event $w_g$, which can be expressed as

$$EDNS = \sum_{i=1}^{N_s} (D_{i,L}^0 - D_{i,L}^{w_g}) \tag{14}$$

where $(D_{i,L}^0 - D_{i,L}^{w_g})$ indicates the amount of electrical load lost at node $i$ when the gas supply fluctuation event $w_g$ occurs; $N_s$ is the number of loads. The larger the *EDNS*, the larger the electrical load removed due to the fault and the lower the system reliability.

Severity Index *SI*: Based on the *EDNS* obtained from (14), calculating the ratio of the electric load removed from the system at the load point due to a gas supply fluctuation event $w_g$ (e.g., due to policy regulation or economic fluctuations) to the total electrical load that would have occurred if no gas supply loss event had occurred, which can be expressed as

$$SI = \frac{EDNS}{L_{\text{total}}} \times T \tag{15}$$

where $L_{\text{total}}$ is the total system electrical load and $T$ is the time from the gas supply loss event to repair. *SI* represents the system severity, and the larger this indicator is, the greater the severity of the system failure and the lower the system reliability.

Average Service Availability Index *ASAI*: Indicating the ratio of the normal electrical load of the system in this scenario to the total electrical load when no gas loss event occurs [42], which can be expressed as

$$ASAI = 1 - \frac{EDNS}{L_{\text{total}}} \tag{16}$$

where *ASAI* is the power supply availability of the system. The larger the value, the higher the power supply margin of the system and the higher the system reliability.

Average Energy Not Supplied *AENS*: Indicating the average amount of power shortage at each node when a load-shedding event occurs, which can be expressed as

$$AENS = \frac{EDNS}{N_s} \tag{17}$$

where *AENS* is the average amount of load loss per node in a power system. The more severe the power system failure, the lower the system reliability.

## 4. Case Studies

### 4.1. Natural Gas Supply Chain and Power System Initial Parameter Setting

A natural gas system with a 14-node power system of a certain area [43], as shown in Figure 4, is used to illustrate the effectiveness of the proposed model. The stations in the natural gas pipeline network system are categorized into pressure stations, sub-transmission stations, and pressure sub-transmission stations [44]. These station types are represented topologically using squares, circles, and hexagons, respectively.

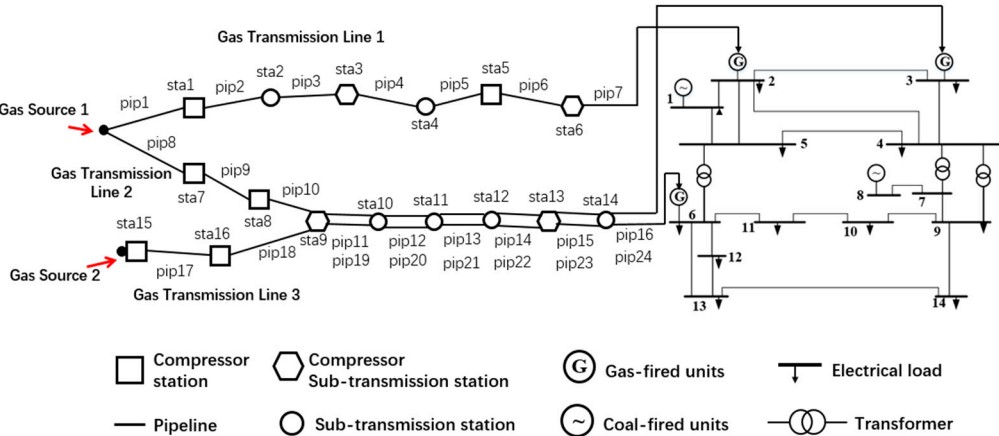

**Figure 4.** Topology diagram of the natural gas system with a 14-node power system of a certain area.

In the natural gas pipeline network system, each line can be viewed as a series of stations and pipelines that can be operated independently. The system consists of three gas-transmission lines, with the second and third lines sharing some station and pipeline sections. It also includes two natural gas sources (red arrows represent the directions of gas flow), six compressor stations, four compressor sub-transmission stations, six sub-transmission stations, and 24 gas-transmission pipelines.

The search tree [15,16] shown in Figure 5 is then sought, with line 1 represented by box A, the independent pipeline segment of line 2 by B, the common pipeline segment with line 3 by D, the independent pipeline segment of line 3 by C, and the common pipeline segment with line 2 by E. The number 1 represents the gas sources, number 2,3 represent the sharing station (sta9), number 4 represents the load side (i.e., the 14-node power system).

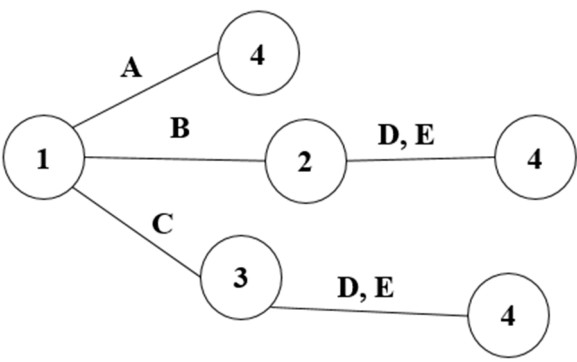

**Figure 5.** Simplified topology diagram of natural gas pipe network system and the power grid of an area.

The maximum volume of gas delivered by line *A* of cut set 1 is $C_{1,A}$, and the working volume is $S_{1,A}$. Similarly, the maximum and working volumes of the other lines can be expressed. A margin of $h\%$ is set for each line ($h = 10$ for this paper) [45]. The values of gas delivery for each line are set in Table 1, and the failure parameters of each component of the natural gas pipeline system in the initial state are set in Table 2 [46].

**Table 1.** The gas-transmission parameters of the line.

| Line Name | Maximum Gas Delivery Volume/m³ | Gas Delivery Volume during Operation/m³ |
|---|---|---|
| A | 5000 | 4500 |
| B | 6000 | 5400 |
| C | 7000 | 6300 |
| D | 6000 | 5400 |
| E | 7000 | 6300 |

**Table 2.** Fault parameters of components in the initial natural gas pipe network system.

| Equipment Components | Failure Probability |
|---|---|
| Compressor station | 0.14 |
| Sub-transmission substation | 0.20 |
| Compressor and sub-transmission station | 0.15 |
| Independent pipeline | 0.10 |
| Collinear pipeline | 0.13 |

In this case, the key factors are set as policy regulations and economic market fluctuations [47]. The amount of natural gas delivered to each gas-fired unit by the three transmission lines is obtained after 10,000 Monte Carlo simulations, and the output and upper and lower limits of each generating unit's output are modified accordingly. Table 3 shows that the units at nodes 1 and 8 of the power grid of an area correspond to coal-fired units, while the units at nodes 2, 3, and 6 correspond to gas-fired units. In this paper, the flow rate of the transmission line in the initial state is used as the benchmark, and 1.5 times the power flowing through each transmission line is used as the upper limit of the transmission capacity of the line.

**Table 3.** Initial operation mode of the power system.

| Nodes | Gas Unit Output/MW | Coal-Fired Unit Output/MW | The Upper Limit of Gas Unit Output/MW | The Upper Limit of Coal-Fired Unit Output/MW | Load/MW |
|---|---|---|---|---|---|
| 1 | \ | 100 | \ | 140 | 0 |
| 2 | 8.8 | \ | 30 | \ | 21.7 |
| 3 | 4 | \ | 50 | \ | 94.2 |
| 6 | 6.73 | \ | 20 | \ | 11.2 |
| 8 | \ | 90 | \ | 100 | 0 |

*4.2. Natural Gas Supply Chain Minimum Cut Set*

Considering the correlation between the reliability of the power system and the natural gas system, it is necessary to calculate the final gas supply to the power system by considering the natural gas delivered from the gas source through the pipeline and the gas-processing station, which determines the final output of the gas-fired power units. To obtain the final gas supply, it is necessary to calculate the possible gas delivery paths, also known as the minimum cut set, from the gas source to the gas-fired power units. Based on Section 2.1, the concatenated set matrix $T$ of Figure 5 is obtained as:

$$T = \begin{bmatrix} 1 & 0 & 0 & 0 & 0 \\ 0 & 1 & 0 & 1 & 0 \\ 0 & 1 & 0 & 0 & 1 \\ 0 & 0 & 1 & 1 & 0 \\ 0 & 0 & 1 & 0 & 1 \end{bmatrix}$$

where the columns are numbered $A$, $B$, $C$, $D$, and $E$. It is known that the system consists of two third-order minimal cut sets, $(A, B, C)$ and $(A, D, E)$, and the system can be further represented as a series of minimal cut sets.

*4.3. Analysis of Power Balance of the Power System Considering the Impact of Gas Supply Fluctuations*

According to the analysis of the natural gas system, key factors impact the supply of natural gas, which further affects the transmission lines of the power system. From a fairness perspective, it is assumed that the impact generated by policy regulations and economic market fluctuations is proportional for all gas sources. The gas supply is used as input to the power system, and the amount of power-load loss under different gas supply levels is calculated and presented in Table 4. To present the data in Table 4 more clearly, nodes 2, 3, and 6 where gas-fired units are connected are selected for plotting, to analyze the amount of power-load loss under different gas supply volumes, which is shown in Figure 6.

**Table 4.** Electric load loss under different gas supplies using Monte Carlo simulation.

| Load Number | Electric Load Loss at 100% Supply/MW | Electric Load Loss at 80% Supply/MW | Electric Load Loss at 60% Supply/MW | Electric Load Loss at 40% Supply/MW | Electric Load Loss at 20% Supply/MW | Electric Load Loss with No Supply/MW |
|---|---|---|---|---|---|---|
| 1 | 0 | 0 | 0 | 0 | 0 | 0 |
| 2 | 0 | 4.47 | 4.79 | 5.12 | 5.45 | 5.78 |
| 3 | 0 | 19.40 | 20.80 | 22.24 | 23.67 | 25.10 |
| 4 | 0 | 9.85 | 10.55 | 11.28 | 12.01 | 12.73 |
| 5 | 0 | 1.57 | 1.68 | 1.79 | 1.91 | 2.02 |
| 6 | 0 | 2.31 | 2.47 | 2.64 | 2.81 | 2.98 |
| 7 | 0 | 0 | 0 | 0 | 0 | 0 |
| 8 | 0 | 0 | 0 | 0 | 0 | 0 |
| 9 | 0 | 6.08 | 6.51 | 6.96 | 7.41 | 7.86 |
| 10 | 0 | 1.85 | 1.99 | 2.12 | 2.26 | 2.40 |
| 11 | 0 | 0.72 | 0.77 | 0.83 | 0.88 | 0.93 |
| 12 | 0 | 1.26 | 1.35 | 1.44 | 1.53 | 1.63 |
| 13 | 0 | 2.78 | 2.98 | 3.19 | 3.39 | 3.60 |
| 14 | 0 | 3.07 | 3.29 | 3.52 | 3.74 | 3.97 |

As can be seen from Table 4 and Figure 6, gas supply curtailment results in an increasing amount of power load being removed. This is because the reduction in the gas supply will lead to insufficient spare capacity of the power system, and in the case that the output of coal-fired units cannot be increased continuously, the power load removed proportionally is needed to maintain the power balance of the system. It is worth mentioning that no

loss of load occurs because coal-fired units made up for the shortage of gas-fired units at load nodes 1, 7, and 8.

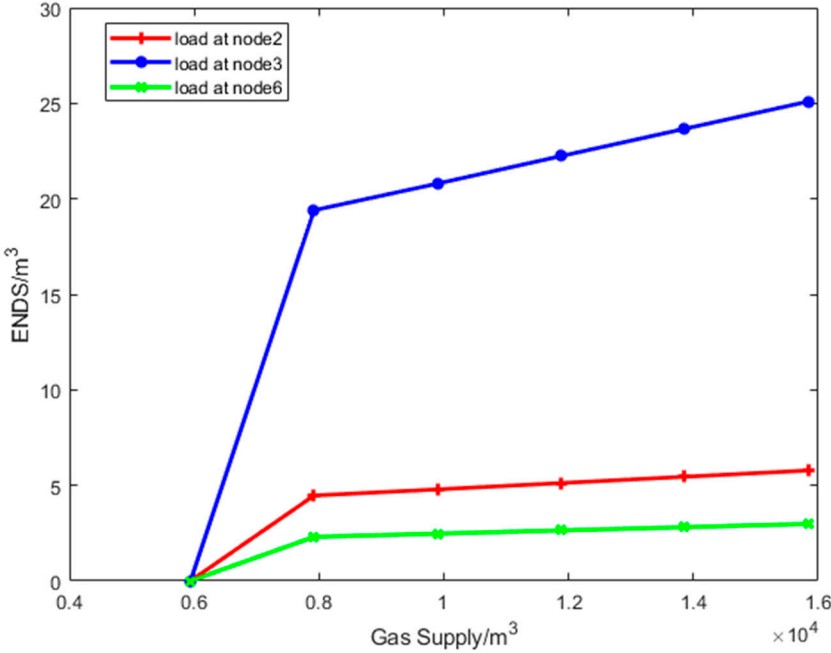

**Figure 6.** Electric load loss under the different gas supplies (loads at nodes 2, 3 and 6).

*4.4. Power System Reliability Evaluation*

The reliability indexes of the power system at different gas supply volumes are calculated through Monte Carlo simulation and presented in Table 5. To prove the correctness of the model developed, the Monte Carlo sampling method (MCS) and Latin hypercube sampling (LHS) method are applied to simulate the natural gas-transmission-loss event (when the gas supply volume is 10,800 $m^3$) and obtain the corresponding reliability-assessment results of the power system, respectively, which is presented in Table 6.

**Table 5.** Reliability indexes under different gas supplies using Monte Carlo simulation.

| Gas Supply Volume/m³ | EDNS/MW | SI | ASAI | AENS |
|---|---|---|---|---|
| 0 | 69.00 | 0.27 | 0.73 | 4.93 |
| 3600 | 65.07 | 0.25 | 0.75 | 4.65 |
| 7200 | 61.15 | 0.24 | 0.76 | 4.37 |
| 10,800 | 57.19 | 0.22 | 0.78 | 4.09 |
| 14,400 | 53.35 | 0.21 | 0.79 | 3.81 |
| 18,000 | 0.00 | 0.00 | 1.00 | 0.00 |

**Table 6.** Comparison of reliability-assessment results using different sampling methods.

| | EDNS/MW | SI | ASAI | AENS |
|---|---|---|---|---|
| **MCS** | 57.19 | 0.22 | 0.78 | 4.09 |
| **LHS** | 57.48 | 0.22 | 0.78 | 4.11 |

The results in Table 5 show that when there is an equal reduction in the gas supply volume, *EDNS* increases, leading to an increase in the power-load removal due to the fault. When the gas supply volume equals 18,000 $m^3$, which means no gas supply loss event occurs, no power-load loss in the power system, *EDNS*, *SI*, and *AENS* equal 0, and *ASAI* equals 1. Then when the gas supply is 14,400 $m^3$, that is, the gas supply is 80% of the original, *EDNS* equals 53.35 MW, *SI* increases to 0.21, *ASAI* decreases to 0.79, and *AENS*

increases to 3.81. A gas supply reduction leads to an increase in *SI* and *AENS*, indicating that the severity of system faults gradually increases. Additionally, *ASAI* decreases, indicating that the power supply availability gradually decreases. These results demonstrate that the reliability of the power system decreases with the gas supply of the gas source, and the power system may not have enough energy to ensure the power balance of electricity.

Table 6 shows that by using both Monte Carlo sampling (MCS) and Latin hypercube sampling (LHS) to simulate power system faults caused by fluctuations in the natural gas supply, the reliability-assessment results of the power system are similar. The *EDNS* obtained with the two methods is 57.19 MW and 57.48 MW, respectively, with only a slight difference of 0.52%. For the results of Average Energy Not Supplied (*AENS*), there is only a negligible difference of 0.47% between the results obtained with the two sampling methods. The results of the Severity Index (*SI*) and Average Service Availability Index (*ASAI*) are consistent, indicating that the model proposed in this paper has a certain degree of universality under different algorithmic solutions.

## 5. Conclusions

For regions with high loads and a high proportion of gas-fired units, the transmission loss of natural gas can affect the reliable power supply. To address this issue, a reliability-evaluation method for the power system considering natural gas supply fluctuations is proposed. Firstly, a reliability model of the natural gas supply chain based on the minimal cut set theory is constructed and the influence of policy regulations and economic market factors on system components is quantified. Secondly, a reliability-evaluation model for a power system that considers gas loss is constructed, and a non-sequential Monte Carlo simulation is used to solve it. Thirdly, a reliability-evaluation method considering the power system reserve capacity is proposed. Using the method to deal with the gas–electric system of a certain area, the main conclusions are summarized as follows:

(1) The reliability of the natural gas supply is influenced by macro factors such as policy regulations and economic market fluctuations. The proactive promotion of low-carbon energy transformation policies will increase the proportion of natural gas, leading to further increases in the natural gas supply and demand while changes in the international situation, such as the Ukrainian crisis, have greatly altered the natural gas trade pattern. The same occurs for electricity prices, which have to some extent affected the natural gas supply. They can lead to a reduction in the gas supply from the source, thereby depriving the power system with a high proportion of gas-fired units of sufficient natural gas for power generation.

(2) Coal-fired units can make up for the shortage of gas-fired units in areas with a high proportion of gas-fired units. In the case study, coal-fired units are connected at nodes 1, 7, and 8. From the results in Table 4, it can be seen that under different gas supply levels, there is no load shedding at nodes 1, 7, and 8, indicating that the coal-fired units are able to compensate for the insufficient output of gas-fired units, keep a balance of electric power at these nodes, and ensure the secure and reliable operation of the power system.

(3) The reliability of the gas supply chain directly affects the reliability of the power system. When external macro factors such as policy regulation and economic fluctuations have an impact on the natural gas supply, assuming a proportional effect on all natural gas sources, as shown in Table 5, when the gas supply decreases from 18,000 m$^3$ to 7200 m$^3$, *EDNS* increases to 61.15 MW, *SI* increases to 0.24, *ASAI* decreases to 0.76, and *AENS* increases to 4.37. When the negative impact increases to the point where the gas supply drops to zero, *EDNS* reaches 69 MW, *SI* increases to 0.27, *ASAI* decreases to 0.73, and *AENS* increases to 4.93, indicating that the severity of system faults is extremely high and the availability of power supply in the electric power system also greatly decreases.

(4) The reliability assessment of power systems considering the natural gas loss method proposed has a certain generality. As shown in Table 6, by using both Monte Carlo sampling (MCS) and Latin hypercube sampling (LHS) to simulate power system faults caused by fluctuations in the natural gas supply, the reliability-assessment results of the power system

are similar. The *EDNS* obtained using the two methods are 57.19 MW and 57.48 MW, respectively, with only a slight difference of 0.52%. For the results of Average Energy Not Supplied (*AENS*), there is only a negligible difference of 0.47% between the results obtained by the two sampling methods. The results of the Severity Index (*SI*) and Average Service Availability Index (*ASAI*) are consistent, indicating that the model proposed in this paper has a certain degree of universality under different algorithmic solutions.

To sum up, the proposed method provides a decision-making basis for the planning and operation of the power system, it also demonstrates that the reliability of the energy supply directly affects the reliability of the power system.

## 6. Future Directions

(1) The demand side of the natural gas supply chain is not fully discussed, such as the interaction between the electricity system and other gas loads. Therefore, the next stage of this study could consider the interaction between gas loads and evaluate the reliability of the electricity system.

(2) The key factors affecting the natural gas supply chain's gas-delivery task are not only policy regulation and market economic fluctuation but also the specific policies and market environment changes that will have varying degrees of impact on natural gas. This paper did not differentiate each factor, and in the future, the impact of key factors' rapid changes on the natural gas supply chain's critical links and their effects on evaluating the reliability of the power system caused by natural gas supply fluctuations will be considered comprehensively.

(3) The response of the power system to gas supply fluctuations has not been considered yet, and there is also storage equipment along the pipeline that needs to be taken into account. How to properly consider user behavior and energy storage equipment is also a next direction of this research.

**Author Contributions:** Conceptualization, K.Z., Y.W., S.Y., X.J., Y.M., J.M. and Z.L.; methodology, K.Z., Z.L. and Y.W.; software, K.Z. and X.J.; validation, K.Z., Y.M. and Y.W.; formal analysis, K.Z., Y.W., S.Y. and Z.L.; investigation, K.Z., S.Y. and J.M.; resources, K.Z., Y.M., J.M. and Z.L.; data curation, S.Y., X.J. and Y.M.; writing—original draft preparation, K.Z. and Y.W.; writing—review and editing, K.Z., Y.W., S.Y., Y.M., J.M. and Z.L.; visualization, K.Z. and J.M.; supervision, Y.W., X.J., Y.M., J.M. and Z.L. All authors have read and agreed to the published version of the manuscript.

**Funding:** This work was supported by the National Natural Science Foundation of China (National Natural Science Foundation of China, 52077195).

**Institutional Review Board Statement:** Not applicable.

**Informed Consent Statement:** Not applicable.

**Data Availability Statement:** The data presented in this study are available on request from the first author.

**Conflicts of Interest:** The authors declare no conflict of interest.

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
