# Peer review of "Reliability Assessment of Power Systems in High-Load Areas with High Proportion of Gas-Fired Units Considering Natural Gas Loss"

_applsci, doi:10.3390/app13106012_

Round 1

Reviewer 1 Report

The article contains interesting research results of the electric load loss under the different gas supply, which may contribute to better transmission. In addition, the obtained results of the values: Severity index and System Availability indicate the essence of the issue, which still requires further research. Below ere are some minor comments and suggestions:

1. In the introduction, it is necessary to add information about the percentage of gas in energy production compared to other raw materials - this is quite interesting information due to the huge extraction of hard coal at the level of 4.5 billion tons;

2. In the subsection 2.1, it should be written how many times the iteration for the matrix T is performed - is there any limit value set;

3. In the subsection 3.1, it should be stated more clearly why the Monte Carlo method was chosen;

4. For equation 15, it should specified what are the causes of fault occurrence;

5. For Figure 4, it should be added explanations to numbers 1 to 14 (right part of the figure);

6. For the data presented in Table 4, a chart should be made with individual ranges marked - this will significantly improve the readability of the results and their meaning;

7. In the fourth chapter, several literature items should be added in which the gas transmission parameters of the line and electric load loss under the different gas supply were tested - so that the results could be compared with other studies and that they would constitute a form of discussion;

8. In the conclusions, numerical values resulting from the calculations presented in the fourth chapter should be added.

Author Response

Dear Reviewer,

       Thank you very much for your constructive comments, which have assisted us in improving the quality of this paper. The paper has been thoroughly revised based on your and the other reviewers’ suggestions. The main modifications for which are outlined in the following and have been marked in red in the manuscript. Our specific comments are detailed as follows. (Please see the attachment)

Reviewer 2 Report

The subject of the article entitled Reliability Assessment of Power Systems Considering Natural Gas Loss, is utterly interesting.

Some issues that need of attendance are as following:

1.       The authors should further elaborate the introduction section in order to justify the contribution of this work as well point to some new aspects on this issue since in literature several articles considering this subject of research can be found.

2.       The methodology which is followed in this work is considered to be adequately defined and validated. Nevertheless, the assessment of the validation results should be further discussed and presented in a separate section in the article so as to permit other researchers to reproduce certain aspects of the research outcomes. The results must be thoroughly interpreted in perspective of the working hypotheses, and the findings of the research as well as to their implications in the broadest context possible.

3.       The final section of the article is rather brief. The conclusions of the research and their association with the results need to be further defined. Moreover, some details about future directions should be pointed point out.

4.       The paper is well-structured in general and written in appropriate English language according to the standards of the Journal, however some spell-checking is required.

Minor editing is required.

Author Response

(The authors gave the same response as above.)

Reviewer 3 Report

To start with, I would like to thank authors for their work in terms of interesting topic and well written article.

The paper is devoted to the development of a reliability evaluation method for power systems containing gas and coal-fired energy sources.

Reviewing the paper, I can highlight the following strengths:

·         The theme of article is in the scope of SI topic “New Insights in Power System Operations and Planning”.

·         There are all essential sections

·         Figures are clear

·         Used English is at good level, text is easy to read.

·          References are sufficient and up-to-date, no impropriate self-citation is detected

The following problems:

1.     The novelty is not clear. Evaluation of power supply system reliability with natural gas sources are discussed in many papers. For example, the first found by  Google - DOI: 10.1109/PTC.2003.1304696

2.     Abstract, delete listing. (First, second etc)

3.     Introduction is cohesive and coherent. Why does it start with the situation in China while the paper aimed at the global problem? What is the reason sharp listing references. Deep reference analysis is essential. Novelty should be shown better.

4.     From the model is not clear how it considers policy regulations and economic market fluctuations. The fluctuation can be caused by many other reasons including faults.

5.     Where are calls out to figures 1 and 2?

6.     I recommend to use the global indexes from  IEEE, for example, ASIDI ( see IEEE Std 1366-2012)

7.     What is the 14-node power system used for a case study? Is it a real object? Where failure probabilities are taken from ?

8.     Why the power supply system reliability model contains only gas and coal-fired energy sources? What about other sources? Abstract and title is not appropriate due to this reason.

9.     Are there any existing power supply system reliability models considering gas sources. How to check the correctness of your method? Discussion section need improving

10.  “The main conclusions” presented in conclusion section is already well known. It needs revising.

Used English is at a good  level, text is easy to read.

Author Response

(The authors gave the same response as above.)

Round 2

Reviewer 3 Report

Dear authors,

Thank you for the paper revision and deep answers to my concerns.

I have some concerns to contribution to the field, but  more or less satisfied with your answers.

Please, correct a  sense mistake:

"Sanctions on Russia have reshaped the crude oil trading pattern, forcing Russia's crude oil exports to shift from the East to the West." ... to shift from the West to the East."

English is much better, at a good level

Author Response

Dear Reviewer,

       Thank you very much for your constructive comments, which have assisted us in improving the quality of this paper. The paper has been thoroughly revised based on your and the other reviewers’ suggestions. The main modifications for which are outlined in the following and have been marked in red in the manuscript. Our specific comments are detailed as follows.
